# Hybrid Memoised Wake-Sleep: Approximate Inference at the Discrete-Continuous Interface

Tuan Anh Le[1]    Katherine M. Collins[1]    Luke Hewitt[1]    Kevin Ellis[2]

N. Siddharth[3]    Samuel Gershman[4]    Joshua B. Tenenbaum[1]

## Abstract

Modeling complex phenomena typically involves the use of both discrete and continuous variables. Such a setting applies across a wide range of problems, from identifying trends in time-series data to performing effective compositional scene understanding in images. Here, we propose *Hybrid Memoised Wake-Sleep* (HMWS), an algorithm for effective inference in such hybrid discrete-continuous models. Prior approaches to learning suffer as they need to perform repeated expensive inner-loop discrete inference. We build on a recent approach, Memoised Wake-Sleep (MWS), which alleviates part of the problem by memoising discrete variables, and extend it to allow for a principled and effective way to handle continuous variables by learning a separate recognition model used for importance-sampling based approximate inference and marginalization. We evaluate HMWS in the GP-kernel learning and 3D scene understanding domains, and show that it outperforms current state-of-the-art inference methods.

## 1 Introduction

We naturally understand the world around us in terms of discrete symbols. When looking at a scene, we automatically parse what is where, and understand the relationships between objects in the scene. We understand that there is a book and a table, and that the book is on the table. Such symbolic representations are often necessary for planning, communication and abstract reasoning. They allow the specification of goals like "book on shelf" or preconditions like the fact that "book in hand" must come before "move hand to shelf", both of which are necessary for high-level planning. In communication, we're forced to describe such plans, among many other things we say, in discrete words. And further, abstract reasoning requires creating new symbols by composing old ones which allows us to generalize to completely new situations. A "tower" structure made out of books is still understood as a "tower" when built out of Jenga blocks. How do we represent and learn models that support such symbolic reasoning while supporting efficient inference?

Figure 1: Hybrid generative model $p_\theta(\boldsymbol{z}_d, \boldsymbol{z}_c, \boldsymbol{x})$.

We focus on a particular class of *hybrid generative models* $p_\theta(z_d, z_c, x)$ of observed data $x$ with discrete latent variables $z_d$, continuous latent variables $z_c$ and learnable parameters $\theta$ with a graphical model structure shown in Figure 1. In particular, the discrete latent variables $z_d$ represent an underlying structure present in the data, while the remaining continuous latent variables $z_c$ represent non-structural quantities. For example, in the context of compositional scene understanding, $z_d$ can represent a scene graph comprising object identities and the relationships between them, like "a small green pyramid is on top of a yellow cube; a blue doughnut leans on the yellow cube; the large pyramid is next to the yellow cube" while $z_c$ represents the continuous poses of these objects. Here, we assume that object identities are discrete, symbolic variables indexing into a set of *learnable* primitives, parameterized by a subset of the generative model parameters $\theta$. The idea is to make these primitives learn to represent concrete objects like "yellow cube" or "large green pyramid" from data in an unsupervised fashion.

Algorithms suitable for learning such models are based on variational inference or wake-sleep. However, these algorithms are either inefficient or inapplicable to general settings. First, stochastic variational inference methods that optimize the evidence lower bound (ELBO) using the reparameterization trick (Kingma & Welling, 2014; Rezende & Mohamed, 2015) are not applicable to discrete

---

[1]MIT, [2]Cornell University, [3]University of Edinburgh & The Alan Turing Institute, [4]Harvard University

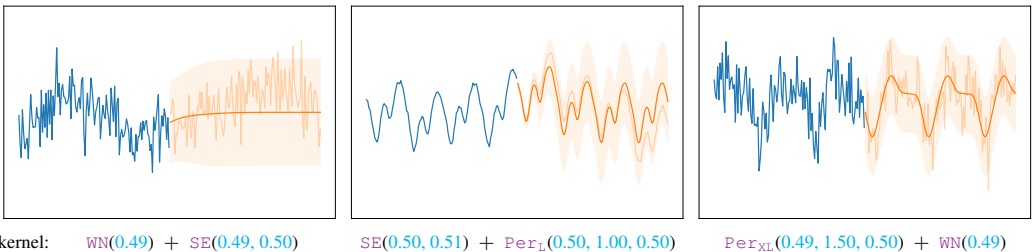

kernel:     WN(0.49) + SE(0.49, 0.50)        SE(0.50, 0.51) + Per$_L$(0.50, 1.00, 0.50)        Per$_{XL}$(0.49, 1.50, 0.50) + WN(0.49)

(a) Trends in time-series: Learning Gaussian process (GP) kernel to fit data (blue). Extrapolation (orange) shown with inferred kernel expression (below).

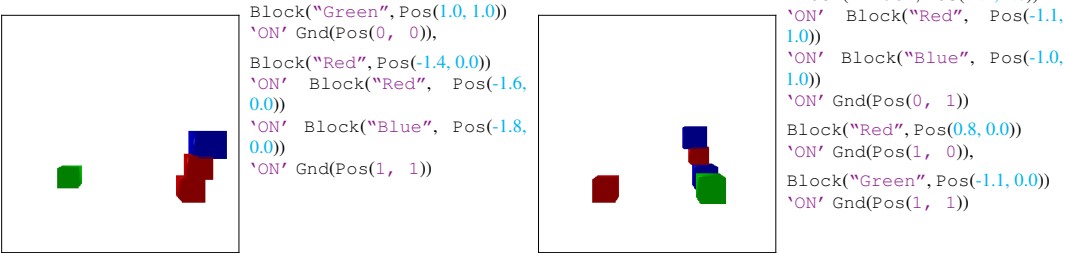

(b) Compositional scene understanding over primitives, attributes, and spatial arrangements.

Figure 2: Examples highlighting the hybrid discrete-continuous nature of data. Colours indicate samples from discrete (violet) and continuous (cyan) random variables respectively.

latent variable models. The REINFORCE gradient estimator (Williams, 1992), on the other hand, has high variance and continuous relaxations of discrete variables (Jang et al., 2017; Maddison et al., 2017) don't naturally apply to stochastic control flow models (Le et al., 2019). Second, wake-sleep methods (Hinton et al., 1995; Dayan et al., 1995) like reweighted wake-sleep (RWS) (Bornschein & Bengio, 2015; Le et al., 2019) require inferring discrete latent variables at every learning iteration, without saving previously performed inferences. Memoised wake-sleep (MWS) (Hewitt et al., 2020) addresses this issue, but is only applicable to purely discrete latent variable models.

We propose hybrid memoised wake-sleep (HMWS)—a method for learning and amortized inference in probabilistic generative models with hybrid discrete-continuous latent variables. HMWS combines the strengths of MWS in memoising the discrete latent variables and RWS in handling continuous latent variables. The core idea in HMWS is to memoise discrete latent variables $z_d$ and learn a separate recognition model which is used for importance-sampling based approximate inference and marginalization of the continuous latent variables $z_c$.

We empirically compare HMWS with state-of-the-art (i) stochastic variational inference methods that use control variates to reduce the REINFORCE gradient variance, VIMCO (Mnih & Rezende, 2016), and (ii) a wake-sleep extension, RWS. We show that HMWS outperforms these baselines in two domains: structured time series and compositional 3D scene understanding, respectively.

## 2 BACKGROUND

Our goal is to learn the parameters $\theta$ of a generative model $p_\theta(z, x)$ of latent variables $z$ and data $x$, and parameters $\phi$ of a recognition model $q_\phi(z|x)$ which acts as an approximation to the posterior $p_\theta(z|x)$. This can be achieved by maximizing the evidence lower bound

$$\text{ELBO}\left(x, p_\theta(z, x), q_\phi(z|x)\right) = \log p_\theta(x) - \text{KL}(q_\phi(z|x)||p_\theta(z|x)) \quad (1)$$

which maximizes the evidence $\log p_\theta(x)$ while minimizing the Kullback-Leibler (KL) divergence, thereby encouraging the recognition model to approximate the posterior.

If the latent variables are discrete, a standard way to estimate the gradients of the ELBO with respect to the recognition model parameters involves using the REINFORCE (or score function) gradient estimator (Williams, 1992; Schulman et al., 2015)

$$\nabla_\phi \text{ELBO}\left(x, p_\theta(z, x), q_\phi(z|x)\right) \approx \log \frac{p_\theta(z, x)}{q_\phi(z|x)} \cdot \nabla_\phi \log q_\phi(z|x) + \nabla_\phi \log \frac{p_\theta(z, x)}{q_\phi(z|x)} \quad (2)$$

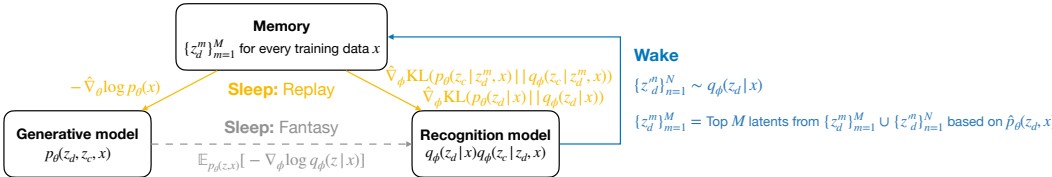

Figure 3: Learning phases of Hybrid Memoised wake-sleep. The memory stores a set of discrete latents for each training data point $x$. The **Wake** phase updates the memory using samples from the recognition model. The **Sleep: Replay** phase updates the generative model and the recognition model using the memory. The **Sleep: Fantasy** phase updates the recognition model using samples from the generative model.

where $z \sim q_\phi(z|x)$. However, the first term is often high-variance which makes learning inefficient. This issue can be addressed by (i) introducing control variates (Mnih & Gregor, 2014; Mnih & Rezende, 2016; Tucker et al., 2017; Grathwohl et al., 2018) which reduce gradient variance, (ii) continuous-relaxation of discrete latent variables to allow differentiation (Jang et al., 2017; Maddison et al., 2017), or (iii) introducing a separate "wake-sleep" objective for learning the recognition model that is trained in different steps and sidesteps the need to differentiate through discrete latent variables (Hinton et al., 1995; Dayan et al., 1995; Bornschein & Bengio, 2015; Le et al., 2019).

## 2.1 MEMOISED WAKE-SLEEP

Both ELBO-based and wake-sleep based approaches to learning require re-solving the inference task by sampling from the recognition model $q_\phi(z|x)$ at every iteration. This repeated sampling can be wasteful, especially when only a few latent configurations explain the data well. Memoised wake-sleep (MWS) (Hewitt et al., 2020) extends wake-sleep by introducing a *memory*—a set of $M$ unique discrete latent variables $\{z_d^m\}_{m=1}^M$ for each data point $x$ which induces a variational distribution

$$q_{\text{MEM}}(z_d|x) = \sum_{m=1}^M \omega_m \delta_{z_d^m}(z_d), \qquad (3)$$

consisting of a weighted set of delta masses $\delta_{z_d^m}$ centered on the memory elements (see also Saeedi et al. (2017)). This variational distribution is proven to improve the evidence lower bound $\text{ELBO}(x, p_\theta(z_d, x), q_{\text{MEM}}(z_d|x))$ (see Sec. 3 of (Hewitt et al., 2020)) by a memory-update phase comprising (i) the proposal of $N$ new values $\{z'^n_d\}_{n=1}^N \sim q_\phi(z_d|x)$, (ii) retaining the best $M$ values from the union of the old memory elements and newly proposed values $\{z_d^m\}_{m=1}^M \cup \{z'^n_d\}_{n=1}^N$ scored by $p_\theta(z_d, x)$, and (iii) setting the weights to $\omega_m = p_\theta(z_d^m, x)/\sum_{i=1}^M p_\theta(z_d^i, x)$.

MWS, however, only works on models with purely discrete latent variables. If we try to use the same approach for hybrid discrete-continuous latent variable models, all proposed continuous values are will be unique and the posterior approximation will collapse onto the MAP estimate.

## 2.2 IMPORTANCE SAMPLING BASED APPROXIMATE INFERENCE AND MARGINALIZATION

In our proposed method, we will rely on importance sampling (IS) to perform approximate inference and marginalization. In general, given an unnormalized density $\gamma(z)$, and its corresponding normalizing constant $Z = \int \gamma(z) \, dz$ and normalized density $\pi(z) = \gamma(z)/Z$, we want to estimate $Z$ and the expectation of an arbitrary function $f$, $\mathbb{E}_{\pi(z)}[f(z)]$. To do this, we sample $K$ values $\{z_k\}_{k=1}^K$ from a proposal distribution $\rho(z)$, and weight each sample by $w_k = \frac{\gamma(z_k)}{\rho(z_k)}$, leading to the estimators

$$Z \approx \frac{1}{K} \sum_{k=1}^K w_k =: \hat{Z}, \qquad \mathbb{E}_{\pi(z)}[f(z)] \approx \sum_{k=1}^K \bar{w}_k f(z_k) =: \hat{I}, \qquad (4)$$

where $\bar{w}_k = w_k/(K\hat{Z})$ is the normalized weight. The estimator $\hat{Z}$ is often used to estimate marginal distributions, for example $p(x)$, with $\gamma(z)$ being the joint distribution $p(z, x)$. It is unbiased and its variance decreases linearly with $K$. The estimator $\hat{I}$ is often used to estimate the posterior expectation of gradients, for example the "wake-$\phi$" gradient of RWS (Bornschein & Bengio, 2015; Le et al., 2019), $\mathbb{E}_{p_\theta(z|x)}[-\nabla_\phi \log q_\phi(z|x)]$ with $\gamma(z) = p_\theta(z, x)$, $\pi(z) = p_\theta(z|x)$ and $f(z) = -\nabla_\phi \log q_\phi(z|x)$. This estimator is asymptotically unbiased and its asymptotic variance decreases linearly with $K$ (Owen, 2013, Eq. 9.8) which means increasing $K$ improves the estimator.

## 3 HYBRID MEMOISED WAKE-SLEEP

We propose hybrid memoised wake-sleep (HMWS) which extends memoised wake-sleep (MWS) to address the issue of memoization of continuous latent variables. In HMWS, we learn a generative model $p_\theta(z_d, z_c, x)$ of hybrid discrete ($z_d$) and continuous ($z_c$) latent variables, and a recognition model $q_\phi(z_d, z_c|x)$ which factorizes into a discrete recognition model $q_\phi(z_d|x)$ and a continuous recognition model $q_\phi(z_c|z_d, x)$. Like MWS, HMWS maintains a memory of $M$ discrete latent variables $\{z_d^m\}_{m=1}^M$ per data point $x$ which is updated in the *wake* phase of every learning iteration. In the *sleep: replay* phase, we use the memoized discrete latents to train both the generative model and the recognition model. In the *sleep: fantasy* phase, we optionally train the recognition model on data generated from the generative model as well. We summarize these learning phases in Fig. 3, the full algorithm in Alg. 1, and describe each learning phase in detail below. For notational clarity, we present the algorithm for a single training data point $x$.

### 3.1 WAKE

Given a set of $M$ memory elements $\{z_d^m\}_{m=1}^M$, we define the memory-induced variational posterior $q_{\text{MEM}}(z_d|x) = \sum_{m=1}^M \omega_m \delta_{z_d^m}(z_d)$ similarly as in MWS (eq. (3)). If we knew how to evaluate $p_\theta(z_d^m, x)$, the wake-phase for updating the memory of HMWS would be identical to MWS. Here, we use an IS-based estimator of this quantity based on $K$ samples from the continuous recognition model

$$\hat{p}_\theta(z_d^m, x) = \frac{1}{K} \sum_{k=1}^K w_{mk}, \quad w_{mk} = \frac{p_\theta(z_d^m, z_c^{mk}, x)}{q_\phi(z_c^{mk}|z_d^m, x)}, \quad z_c^{mk} \sim q_\phi(z_c|z_d^m, x). \tag{5}$$

The IS weights $\{\{w_{mk}\}_{k=1}^K\}_{m=1}^M$ are used for marginalizing $z_c$ here but they will later be reused for estimating expected gradients by approximating the continuous posterior $p_\theta(z_c|z_d^m, x)$. Given the estimation of $p_\theta(z_d^m, x)$, the weights $\{\omega_m\}_{m=1}^M$ of the memory-induced posterior $q_{\text{MEM}}(z_d|x)$ are

$$\omega_m = \frac{\hat{p}_\theta(z_d^m, x)}{\sum_{i=1}^M \hat{p}_\theta(z_d^i, x)} = \frac{\frac{1}{K}\sum_{k=1}^K w_{mk}}{\sum_{i=1}^M \frac{1}{K}\sum_{j=1}^K w_{ij}} = \frac{\sum_{k=1}^K w_{mk}}{\sum_{i=1}^M \sum_{j=1}^K w_{ij}}. \tag{6}$$

The resulting wake-phase memory update is identical to that of MWS (Sec. 2.1), except that the evaluation of $p_\theta(z_d^m, x)$ and $\omega_m$ is done by (5) and (6) instead. For large $K$, these estimators are more accurate, and if exact, guaranteed to improve $\text{ELBO}(x, p_\theta(z_d, x), q_{\text{MEM}}(z_d|x))$ like in MWS.

### 3.2 SLEEP: REPLAY

In this phase, we use the memoized discrete latent variables $\{z_d^m\}_{m=1}^M$ ("replay from the memory") to learn the generative model $p_\theta(z_d, z_c, x)$, and the discrete and continuous recognition models $q_\phi(z_d|x)$ and $q_\phi(z_c|z_d, x)$. We now derive the gradient estimators used for learning each of these distributions in turn and summarize how they are used within HMWS in the "Sleep: Replay" part of Alg. 1.

---

**Algorithm 1** Hybrid Memoised Wake-Sleep (a single learning iteration)

---

**Input:** Generative model $p_\theta(z_d, z_c, x)$, recognition model $q_\phi(z_d|x)q_\phi(z_c|z_d, x)$, data point $x$, memory elements $\{z_d^m\}_{m=1}^M$, replay factor $\lambda \in [0, 1]$, # of importance samples $K$, # of proposals $N$.
**Output:** Gradient estimators for updating $\theta$ and $\phi$: $g_\theta$ and $g_\phi$; updated memory elements $\{z_d^m\}_{m=1}^M$.

1: Propose $\{z'^n_d\}_{n=1}^N \sim q_\phi(z_d|x)$                                        ▷ **Wake**
2: Select $\{z_d^i\}_{i=1}^L \leftarrow \text{unique}(\{z_d^m\}_{m=1}^M \cup \{z'^n_d\}_{n=1}^N)$ where $L \leq M + N$
3: Sample $\{z_c^{ik}\}_{k=1}^K \sim q_\phi(z_c|z_d^i, x)$ for $i = 1, \ldots, L$
4: Compute $\{\{w_{ik}\}_{k=1}^K\}_{i=1}^L$, $\{\hat{p}_\theta(z_d^i, x)\}_{i=1}^L$ (eq. (5)), and $\{\omega_i\}_{i=1}^L$ (eq. (6))
5: Select $\{z_d^m, \{z_c^{mk}\}_{k=1}^K, \{w_{mk}\}_{k=1}^K, \omega_m\}_{m=1}^M \leftarrow$ best $M$ values from $\{z_d^i\}_{i=1}^L$, $\{\{z_c^{ik}\}_{k=1}^K\}_{i=1}^L$, $\{\{w_{ik}\}_{k=1}^K\}_{i=1}^L$ and $\{\omega_i\}_{i=1}^L$ according to $\{\hat{p}_\theta(z_d^i, x)\}_{i=1}^L$
6: Compute generative model gradient $g_\theta \leftarrow g_\theta(\{w_{mk}, z_c^{mk}\}_{m=1,k=1}^{M,K})$ (eq. (7))        ▷ **Sleep**: Replay
7: Compute $g^\phi_{d,\text{REPLAY}} \leftarrow g^\phi_{d,\text{REPLAY}}(\{z_d^m, \omega_m\}_{m=1}^M)$ (eq. (9))
8: Compute $g^\phi_{c,\text{REPLAY}} \leftarrow g^\phi_{c,\text{REPLAY}}(\{w_{mk}, z_c^{mk}\}_{m=1,k=1}^{M,K})$ (eq. (10))
9: Compute $g^\phi_{\text{FANTASY}} \leftarrow -\frac{1}{K}\sum_{k=1}^K \nabla_\phi \log q_\phi(z_k|x_k)$ from $z_k, x_k \sim p_\theta(z, x)$        ▷ **Sleep**: Fantasy
10: Aggregate recognition model gradient $g_\phi \leftarrow \lambda(g^\phi_{d,\text{REPLAY}} + g^\phi_{c,\text{REPLAY}}) + (1 - \lambda)g^\phi_{\text{FANTASY}}$
11: **Return:** $g_\theta, g_\phi, \{z_d^m\}_{m=1}^M$.

---

**Learning the generative model.** We want to minimize the negative evidence $-\log p_\theta(x)$. We express its gradient using Fisher's identity $\nabla_\theta \log p_\theta(x) = \mathbb{E}_{p_\theta(z|x)}[\nabla_\theta \log p_\theta(z, x)]$ which we estimate using a combination of the memory-based approximation of $p_\theta(z_d|x)$ and the IS-based approximation of $p_\theta(z_c|x)$. The resulting estimator *reuses* the samples $z_c^{mk}$ and weights $w_{mk}$ from the wake phase (see Appendix A for derivation):

$$-\nabla_\theta \log p_\theta(x) \approx -\sum_{m=1}^{M}\sum_{k=1}^{K} v_{mk} \nabla_\theta \log p_\theta(z_d^m, z_c^{mk}, x) =: g_\theta(\{w_{mk}, z_c^{mk}\}_{m=1, k=1}^{M,K}), \quad (7)$$

$$\text{where } v_{mk} = \frac{w_{mk}}{\sum_{i=1}^{M}\sum_{j=1}^{K} w_{ij}}. \quad (8)$$

**Learning the discrete recognition model.** We want to minimize $\text{KL}(p_\theta(z_d|x)||q_\phi(z_d|x))$ whose gradient can be estimated using a memory-based approximation of $p_\theta(z_d|x)$ (eq. (3))

$$\nabla_\phi \text{KL}(p_\theta(z_d|x)||q_\phi(z_d|x)) = \mathbb{E}_{p_\theta(z_d|x)}[-\nabla_\phi \log q_\phi(z_d|x)]$$

$$\approx -\sum_{m=1}^{M} \omega_m \nabla_\phi \log q_\phi(z_d^m|x) =: g_{d,\text{REPLAY}}^\phi(\{z_d^m, \omega_m\}_{m=1}^M), \quad (9)$$

where memory elements $\{z_d^m\}_{m=1}^M$ and weights $\{\omega_m\}_{m=1}^M$ are reused from the wake phase (6).

**Learning the continuous recognition model.** We want to minimize the average of $\text{KL}(p_\theta(z_c|z_d^m, x)||q_\phi(z_c|z_d^m, x))$ over the memory elements $z_d^m$. The gradient of this KL is estimated by an IS-based estimation of $p_\theta(z_c|z_d^m, x)$

$$\frac{1}{M}\sum_{m=1}^{M} \nabla_\phi \text{KL}(p_\theta(z_c|z_d^m, x)||q_\phi(z_c|z_d^m, x)) = \frac{1}{M}\sum_{m=1}^{M} \mathbb{E}_{p_\theta(z_c|z_d^m, x)}[-\nabla_\phi \log q_\phi(z_c|z_d^m, x)]$$

$$\approx -\frac{1}{M}\sum_{m=1}^{M}\sum_{k=1}^{K} \bar{w}_{mk} \nabla_\phi \log q_\phi(z_c^{mk}|z_d^m, x) =: g_{c,\text{REPLAY}}^\phi(\{w_{mk}, z_c^{mk}\}_{m=1,k=1}^{M,K}), \quad (10)$$

where $\bar{w}_{mk} = w_{mk}/\sum_{i=1}^{K} w_{mi}$. Both weights and samples $z_c^{mk}$ are reused from the wake phase (5).

### 3.3 Sleep: Fantasy

This phase is identical to "sleep" in the original wake-sleep algorithm (Hinton et al., 1995; Dayan et al., 1995) which minimizes the Monte Carlo estimation of $\mathbb{E}_{p_\theta(z_d, z_c, x)}[-\nabla_\phi \log q_\phi(z_d, z_c|x)]$, equivalent to training the recognition model on samples from the generative model.

Together, the gradient estimators in the *Wake* and the *Sleep: Replay* and the *Sleep: Fantasy* phases are used to learn the generative model, the recognition model and the memory. We additionally introduce a model-dependent replay factor $\lambda \in [0, 1]$ which modulates the relative importance of the *Sleep: Replay* versus the *Sleep: Fantasy* gradients for the recognition model. This is similar to modulating between the "wake-" and "sleep-" updates of the recognition model in RWS (Bornschein & Bengio, 2015). We provide a high level overview of HMWS in Fig. 3 and its full description in Alg. 1.

**Factorization of the recognition model.** The memory-induced variational posterior $q_{\text{MEM}}(z_d|x)$ is a weighted sum of delta masses on $\{z_d^m\}_{m=1}^M$ which cannot model the conditional posterior $p_\theta(z_d|x, z_c)$ as it would require a continuous index space $z_c$. We therefore use $q_{\text{MEM}}(z_d|x)$ to approximate $p_\theta(z_d|x)$ and $q_\phi(z_c|z_d, x)$ to approximate $p_\theta(z_c|z_d, x)$, which allows capturing all dependencies in the posterior $p_\theta(z_c, z_d|x)$. Relaxing the restriction on the factorization of $q_\phi(z_c, z_d|x)$ is possible, but, we won't be able to model any dependencies of discrete latent variables on preceding continuous latent variables.

## 4 Experiments

In our experiments, we compare HMWS on hybrid generative models against state-of-the-art wake-sleep-based RWS and variational inference based VIMCO. To ensure a fair comparison, we match the number of likelihood evaluations – typically the most time-consuming part of training – across

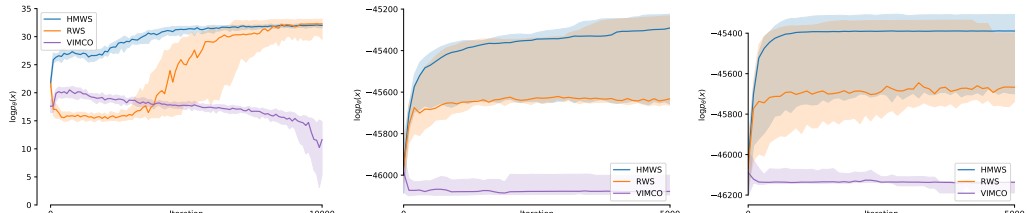

Figure 4: Hybrid memoised wake-sleep (HMWS) learns *faster* than the baselines: reweighted wake-sleep (RWS) and VIMCO based on the marginal likelihood, in both the time series model (left), and the scene understanding models with learning shape and color (middle) and learning shape only (right). HMWS also learns *better* scene understanding models. The gradient estimator of VIMCO was too noisy and failed to learn the time series model. (Median with the shaded interquartile ranges over 20 runs is shown.)

all models: $O(K(N + M))$ for HMWS, and $O(S)$ for RWS and VIMCO where $S$ is the number of particles for RWS and VIMCO. HMWS has a fixed memory cost of $O(M)$ per data point which can be prohibitive for large datasets. However in practice, HMWS is faster and more memory-efficient than the other algorithms as the total number of likelihood evaluations is typically smaller than the upper bound of $K(N + M)$ (see App. C). For training, we use the Adam optimizer with default hyperparameters. We judge the generative-model quality using an $S_{\text{test}} = 100$ sample importance weighted autoencoder (IWAE) estimation of the log marginal likelihood $\log p_\theta(x)$. We also tried the variational-inference based REINFORCE but found that it underperforms VIMCO.

We focus on generative models in which (i) there are interpretable, learnable symbols, (ii) the discrete latent variables represent the composition of these symbols and (iii) the continuous latent variables represent the non-structural quantitites. While our algorithm is suitable for any hybrid generative model, we believe that this particular class of neuro-symbolic models is the most naturally interpretable, which opens up ways to connect to language and symbolic planning.

## 4.1 STRUCTURED TIME SERIES

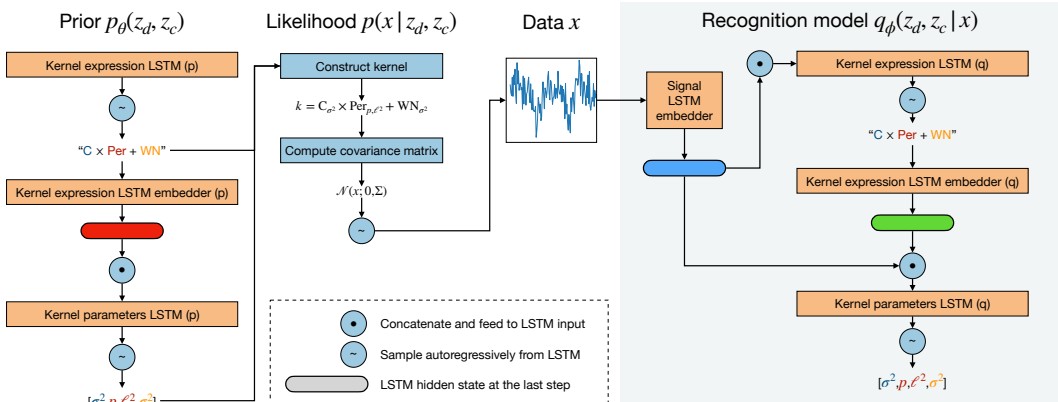

Figure 5: The generative model of structured time series data consists of (i) a prior that uses LSTMs to first sample the discrete Gaussian process (GP) kernel expression and then the continuous kernel parameters, and (ii) a likelihood which constructs the final GP kernel expression. The recognition model (on gray background) mirrors the architecture of the prior but additionally conditions on an LSTM embedding of data.

We first apply HMWS to the task of finding explainable models for time-series data. We draw inspiration from (Duvenaud et al., 2013), who frame this problem as GP kernel learning (Rasmussen & Williams, 2006). They describe a grammar for building kernels compositionally, and demonstrate that inference in this grammar can produce highly interpretable and generalisable descriptions of the structure in a time series. We follow a similar approach, but depart by learning a set of GP kernels jointly for each in time series in a dataset, rather than individually.

For our model, we define the following simple grammar over kernels

$$k \to k + k \mid k \times k \mid \text{WN} \mid \text{SE} \mid \text{PER} \mid \text{C}, \quad \text{where} \tag{11}$$

- WN is the *White Noise* kernel, $\text{WN}_{\sigma^2}(x_1, x_2) = \sigma^2 \mathbb{I}_{x_1 = x_2}$,

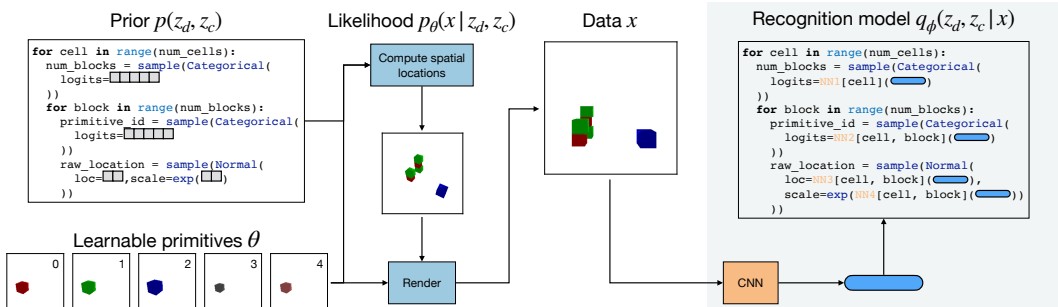

Figure 6: Learning to model structured time series data with Gaussian processes (GPs) by first inferring the kernel expression (shown as text in the top-left corner) and the kernel parameters. The blue curve is the 128-dimensional observed signal, and the orange curve is a GP extrapolation based on the inferred kernel. We show the mean (dark orange), the $\pm 2$ standard deviation range (shaded) and one sample (light orange).

- SE is the *Squared Exponential* kernel, $\text{SE}_{\sigma^2, l^2}(x_1, x_2) = \sigma^2 \exp(-(x_1 - x_2)^2 / 2l^2)$,
- PER is a *Periodic* kernel, $\text{PER}_{\sigma^2, p, l^2}(x_1, x_2) = \sigma^2 \exp(-2 \sin^2(\pi |x_1 - x_2| / p) / l^2)$,
- C is a *Constant*, $\text{C}_{\sigma^2}(x_1, x_2) = \sigma^2$.

We wish to learn a prior distribution over both the symbolic structure of a kernel and its continuous variables ($\sigma^2$, $l$, etc.). To represent the structure of the kernel as $z_d$, we use a symbolic kernel 'expression': a string over the characters $\{(,), +, \times, \text{WN}, \text{SE}, \text{PER}_1, \ldots, \text{PER}_4, \text{C}\}$. We provide multiple character types $\text{PER}_i$ for periodic kernels, corresponding to a learnable coarse-graining of period (short to long). We then define an LSTM prior $p_\theta(z_d)$ over these kernel expressions, alongside a conditional LSTM prior $p_\theta(z_c | z_d)$ over continuous kernel parameters. The likelihood is the marginal GP likelihood—a multivariate Gaussian whose covariance matrix is constructed by evaluating the composite kernel on a fixed set of points. Finally, the recognition model $q_\phi(z_d, z_c | x)$ mirrors the architecture of the prior except that all the LSTMs additionally take as input an LSTM embedding of the observed signal $x$. The architecture is summarized in Fig. 5 and described in full in App. B.1.

We construct a synthetic dataset of 100 timeseries of fixed length of 128 drawn from a probabilistic context free grammar which is constructed by assigning production probabilities to our kernel grammar in (11). In Fig. 4 (left), we show that HMWS learns faster than RWS in this domain in terms of the log evidence $\log p_\theta(x)$. We also trained with VIMCO but due to the high variance of gradients, they were unable to learn the model well.

Examples of latent programs discovered by our model are displayed in Fig. 6. For each signal, we infer the latent kernel expression $z_d$ by taking the highest scoring memory element $z_d^m$ according to the memory-based posterior $q_{\text{MEM}}(z_d |, x)$ and sample the corresponding kernel parameters from the continuous recognition model $q_\phi(z_c | z_d, x)$. We show the composite kernel as well as the GP posterior predictive distribution. These programs describe meaningful compositional structure in the time series data, and can also be used to make highly plausible extrapolations.

## 4.2 COMPOSITIONAL SCENE UNDERSTANDING

Figure 7: Generative model of compositional scenes. The prior places a stochastic number of blocks into each cell, where cells form an imaginary grid on the ground plane. For each cell, a stack of blocks is built by both: i) sampling blocks from a learnable set of primitives and ii) sampling their relative location to the object below (i.e., either the ground or the most recently stacked block). The likelihood uses a differentiable renderer to produce an image. The recognition model (on gray background) mirrors the structure of the prior, but parametrizes distributions as learnable functions (NN1–NN4) of a CNN-based embedding of the image.

Next, we investigate the ability of HMWS to parse images of simple scenes in terms of stacks of toy blocks. Here, towers can be built from a large consortium of blocks, where blocks are drawn from a fixed set of block types. We aim to learn the parameters of the different block types – namely, their size and optionally their color – and jointly infer a scene parse describing which kinds of blocks live

where in each world. Such a symbolic representation of scenes increases interpretability, and has connections to language and symbolic planning.

Our generative and recognition models are illustrated in Fig. 7 (for the full description of the model, see App. B.2). We divide the ground plane into an $N \times N$ ($N = 2$) grid of square cells and initialize a set of $P = 5$ learnable block types, which we refer to as the base set of "primitives" with which the model builds scenes. We parameterize each block type by its size and color. Within each cell, we sample the number of blocks in a tower uniformly, from zero to a maximum number of blocks $B_{\max} = 3$. For each position in the tower, we sample an integer $\in [0, B_{\max}]$, which we use to index into our set of learnable primitives, and for each such primitive, we also sample its continuous position. The (raw) position of the first block in a tower is sampled from a standard Gaussian and is constrained to lie in a subset of the corresponding cell, using an affine transformation over a sigmoid-transformed value. A similar process is used to sample the positions of subsequent blocks, but now, new blocks are constrained to lie in a subset of the top surface of the blocks below.

Given the absolute spatial locations and identities of all blocks, we render a scene using the PyTorch3D differentiable renderer (Ravi et al., 2020), which permits taking gradients of the generative model probability $p_\theta(z_d, z_c, x)$ with respect to the learnable block types $\theta$. All training scenes are rendered from a fixed camera position (a front view); camera position is not explicitly fed into the model. We use an independent Gaussian likelihood with a fixed standard deviation factorized over all pixels – similar to using an $L_2$ loss. The discrete latent variables, $z_d$, comprise both the number of blocks and block type indices, while the continuous latent variables, $z_c$, represent the raw block locations.

The recognition model $q_\phi(z_d, z_c|x)$ follows a similar structure to that of the prior $p(z_d, z_c)$, as shown inside the gray box of Fig. 7. However, the logits of the categorical distributions for $q_\phi(z_d|x)$, as well as the mean and (log) standard deviation of the Gaussian distributions for $q_\phi(z_c|z_d, x)$, are obtained by mapping a convolutional neural network (CNN)-based embedding of the image $x$ through small neural networks NN1–NN4; in our case, these are linear layers.

We train two models: one which infers scene parses from colored blocks and another which reasons over unicolor blocks. In both settings, the model must perform a non-trivial task of inferring a scene parse from an exponential space of $P^{B_{\max} \cdot N^2}$ possible scene parses. Moreover, scenes are replete with uncertainty: towers in the front of the grid often occlude those in the back. Likewise, in the second setting, there is additional uncertainty arising from blocks having the same color. For each model, we generate and train on a dataset of 10k scenes. Both models are trained using HMWS with $K = 5, M = 5, N = 5$, and using comparison algorithms with $S = K(N + M) = 50$.

In Fig. 4 (middle and right), we show that in this domain, HMWS learns faster and better models than RWS ad VIMCO, scored according to the log evidence $\log p_\theta(x)$. We directly compare the wall-clock time taken by HMWS compared to RWS in Table 1, highlighting the comparative efficiency of our algorithm. Further, we uncovered two local minima in our domain; we find that a higher portion of HMWS find the superior minima compared to RWS. In Fig. 8, we show posterior samples from our model. Samples are obtained by taking the highest probability scene parses based on the memory-based posterior approximation $q_{\mathrm{MEM}}(z_d|x)$. These samples illustrate that HMWS-trained models can capture interesting uncertainties over occluded block towers and recover the underlying building blocks of the scenes. For instance, in the colored block domain, we see that the model properly discovers and builds with red, blue, and green cubes of similar sizes to the data.

## 5 RELATED WORK

Our work builds on wake-sleep algorithms (Hinton et al., 1995) and other approaches that jointly train generative/recognition models, particularly modern versions such variational autoencoders (Kingma & Welling, 2014) and reweighted wake-sleep (Bornschein & Bengio, 2015). While variational autoencoders (VAEs) have been employed for object-based scene understanding (e.g. Eslami et al. (2016); Greff et al. (2019)), they attempt to circumvent the issues raised by gradient estimation with discrete latent variables, either by using continuous relaxation (Maddison et al., 2017) or limited use of control variates (Mnih & Rezende, 2016). The wake-sleep family of algorithms avoids the need for such modification and is better suited to models that involve stochastic branching (see Le et al. (2019) for a discussion), and is thus our primary focus here.

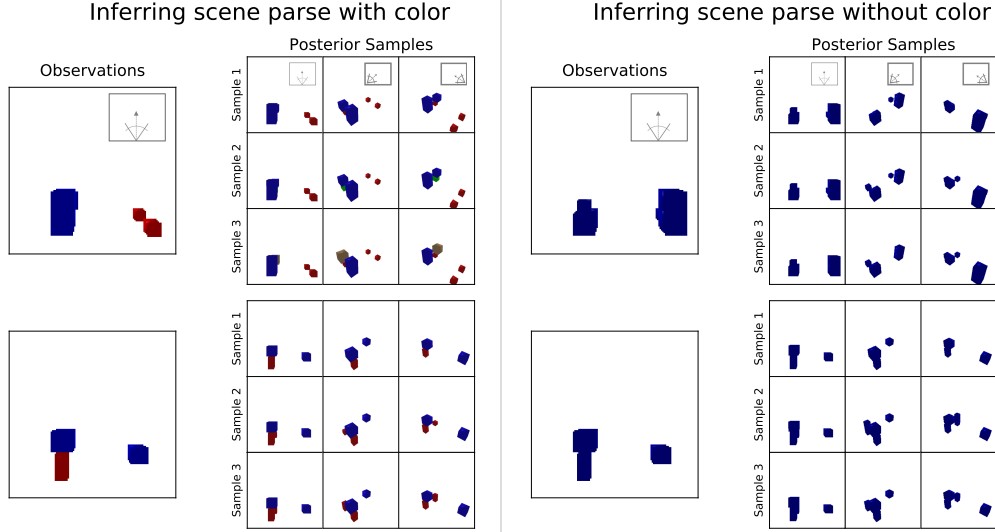

Figure 8: Samples from the posterior when inferring scene parses with color (left) and scene parses without color diversity (right). Conditioned on a single observation of the front view of a scene (left column), HMWS infers a posterior over blocks that make up the scene. Three samples from the posterior are shown per scene, sorted by log probability under the model; e.g., the first sample is most probable under HMWS. Sampled scenes are rendered from three different camera angles; position of the camera is depicted in the figure insets. We emphasize that the model has never seen the 3/4-views. The sampled scenes highlight that we are able to handle occlusion, capturing a distribution over possible worlds that are largely consistent with the observation.

Our most closely related work is Memoised Wake-sleep, which we build directly on top of and extend to handle both discrete and continuous latents. A contemporary of Memoised Wake-sleep, DreamCoder (Ellis et al., 2021), is a closely related program synthesis algorithm following a similar wake-sleep architecture. Like MWS, it lacks principled handling of latent continuous parameters; performing inner-loop gradient descent and heuristically penalizing parameters via the Bayesian Information Criterion (MacKay, 2002). Other instances of wake-sleep models include those that perform likelihood-free inference (Brehmer et al., 2020), or develop more complex training schemes for amortization (Wenliang et al., 2020).

Broadly, our goal of inferring compositionally structured, mixed discrete-continuous objects has strongest parallels in the program synthesis literature. One family of approaches (e.g. HOU-DINI (Valkov et al., 2018) and NEAR (Shah et al., 2020)), perform an outer-loop search over discrete structures and an inner-loop optimization over continuous parameters. Others jointly reason over continuous and discrete parameters via exact symbolic methods (e.g. Evans et al. (2021)) or by relaxing the discrete space to allow gradient-guided optimization to run on the whole problem (e.g. DeepProbLog (Manhaeve et al., 2018)). None of these however, learn-to-learn by amortizing the cost of inference, with recent attempts needing quantized continuous parameters (Ganin et al., 2021). What our work contributes to this space is a generic and principled way of applying amortized inference to hybrid discrete-continuous problems. This gets the speed of a neural recognition model–which is valuable for program synthesis–but with Bayesian handling of uncertainty and continuous parameters, rather than relying on heuristic quantization or expensive inner loops.

## 6  DISCUSSION

Inference in hybrid generative models is important for building interpretable models that generalize. However, such a task is difficult due to the need to perform inference in large, discrete spaces. Unlike deep generative models in which the generative model is less constrained, learning in symbolic models is prone to getting stuck in local optima. While compared to existing algorithms, HMWS improves learning and inference in these models, it does not fully solve the problem. In particular, our algorithm struggles with models in which the continuous latent variables require non-trivial inference, as the quality of continuous inference is directly linked with the quality of the gradient estimators in HMWS. This challenge can potentially be addressed with better gradient estimators. We additionally plan to extend our algorithm to more complex, realistic neuro-symbolic models; for instance, those with more non-cuboidal primitive topologies.

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

## A   DERIVATION OF THE GRADIENT ESTIMATOR FOR LEARNING THE GENERATIVE MODEL

Here, we derive the gradient estimator for learning the generative model given in (7).

We assume that the memory-induced variational posterior approximates the discrete posterior

$$p_\theta(z_d|x) \approx q_{\text{MEM}}(z_d|x) \tag{12}$$

$$= \sum_{m=1}^{M} \omega_m \delta_{z_d^m}(z_d) \tag{13}$$

and that the continuous posterior expectation $\mathbb{E}_{p_\theta(z_c|z_d^m,x)}[f(z_c)]$ for each $m$ is approximated using a set of weighted samples $(z_c^{mk}, \bar{w}_{mk})_{k=1}^K$

$$\mathbb{E}_{p_\theta(z_c|z_d^m,x)}[f(z_c)] \approx \sum_{k=1}^{K} \bar{w}_{mk} f(z_c^{mk}), \tag{14}$$

where $\bar{w}_{mk} = \frac{w_{mk}}{\sum_{i=1}^{K} w_{mi}}$.

First, we make use of the Fisher's identity

$$
\begin{aligned}
\nabla_\theta \log p_\theta(x) &= \mathbb{E}_{p_\theta(z|x)}\left[\nabla_\theta \log p_\theta(x)\right] && \text{(Integrand is not a function of } z) & (15)\\
&= \mathbb{E}_{p_\theta(z|x)}\left[\nabla_\theta \log p_\theta(x) + \nabla_\theta \log p_\theta(z|x)\right] && \text{(Second term vanishes)} & (16)\\
&= \mathbb{E}_{p_\theta(z|x)}\left[\nabla_\theta \log p_\theta(z,x)\right] && \text{(Product rule of probability)} & (17)
\end{aligned}
$$

and we continue by using the approximations in (12)–(14)

$$
\begin{aligned}
&= \mathbb{E}_{p_\theta(z_c|z_d,x)}\left[\mathbb{E}_{p_\theta(z_d|x)}\left[\nabla_\theta \log p_\theta(z,x)\right]\right] && \text{(Factorize the posterior)} & (18)\\
&\approx \mathbb{E}_{p_\theta(z_c|z_d,x)}\left[\mathbb{E}_{q_{\text{MEM}}(z_d|x)}\left[\nabla_\theta \log p_\theta(z,x)\right]\right] && \text{(Use (12))} & (19)\\
&= \mathbb{E}_{p_\theta(z_c|z_d^m,x)}\left[\sum_{m=1}^{M} \omega_m \nabla_\theta \log p_\theta(z_d^m, z_c, x)\right] && \text{(Use (13))} & (20)\\
&\approx \sum_{k=1}^{K} \bar{w}_{mk}\left(\sum_{m=1}^{M} \omega_m \nabla_\theta \log p_\theta(z_d^m, z_c^{mk}, x)\right) && \text{(Use (14))} & (21)\\
&= \sum_{m=1}^{M}\sum_{k=1}^{K} v_{mk} \nabla_\theta \log p_\theta(z_d^m, z_c^{mk}, x), && \text{(Combine sums)} & (22)
\end{aligned}
$$

where

$$v_{mk} = \bar{w}_{mk}\omega_m \tag{23}$$

$$= \frac{w_{mk}}{\sum_{i=1}^{K} w_{mi}} \cdot \frac{\sum_{i=1}^{K} w_{mi}}{\sum_{i=1}^{M}\sum_{j=1}^{K} w_{ij}} \tag{24}$$

$$= \frac{w_{mk}}{\sum_{i=1}^{M}\sum_{j=1}^{K} w_{ij}}. \tag{25}$$

## B   MODEL ARCHITECTURE AND PARAMETER SETTINGS

### B.1   STRUCTURED TIME SERIES

Fixed parameters

- Vocabulary $\mathcal{V} = \{\text{WN}, \text{SE}, \text{PER}_1, \ldots, \text{PER}_4, \text{C}, \times, +, (, )\}$
- Vocabulary size `vocabulary_size` $= |\mathcal{V}| = 11$
- Kernel parameters $\mathcal{K} = \{\sigma_{\text{WN}}^2, (\sigma_{\text{SE}}^2, \ell_{\text{SE}}^2), (\sigma_{\text{PER}_1}^2, p_{\text{PER}_1}, \ell_{\text{PER}_1}^2), \ldots, (\sigma_{\text{PER}_4}^2, p_{\text{PER}_4}, \ell_{\text{PER}_4}^2), \sigma_{\text{C}}^2\}$

- Kernel parameters size `kernel_params_size` = $|\mathcal{K}|$ = 16
- Hidden size `hidden_size` = 128
- Observation embedding size `obs_embedding_size` = 128

Generative model $p_\theta(z_d, z_c, x)$

- Kernel expression LSTM (p)
  (input size = `vocabulary_size`, hidden size = `hidden_size`)
    - This module defines a distribution over the kernel expression $p_\theta(z_d)$.
    - At each time step $t$, the input is a one-hot encoding of the previous symbol in $\mathcal{V}$ (or a zero vector for the first time step)
    - At each time step $t$, extract logit probabilities for each element in $\mathcal{V}$ or the end-of-sequence symbol `EOS` from the hidden state using a "Kernel expression extractor (p)" (`Linear(hidden_size, vocabulary_size + 1)`). This defines the conditional distribution $p_\theta(z_d^t|z_d^{1:t-1})$.
    - The full prior over kernel expression is an autoregressive distribution $\prod_t p_\theta(z_d^t|z_d^{1:t-1})$ whose length is determined by `EOS`.
- Kernel expression LSTM embedder (p)
  (same as kernel expression LSTM (p))
    - The module summarizes the kernel expression $z_d$ into an embedding vector $e \in \mathbb{R}^{\texttt{hidden\_size}}$
    - We re-use the kernel expression LSTM (p) above and use the last hidden LSTM state to be the summary embedding vector.
- Kernel parameters LSTM (p)
  (input size = `kernel_params_size` + `hidden_size`, hidden size = `hidden_size`)
    - This module define a distribution over kernel parameters $p_\theta(z_c|z_d)$.
    - At each timestep $t$, the input is a concatenation of the previous kernel parameters $z_c^{t-1} \in \mathbb{R}^{\texttt{kernel\_params\_size}}$ (or zero vector for $t = 1$) and the embedding vector $e \in \mathbb{R}^{\texttt{hidden\_size}}$.
    - At each timestep $t$, extract mean and standard deviations for each kernel parameter in $\mathcal{K}$ using a "Kernel parameters extractor (p)" `Linear(hidden_size, 2 * kernel_params_size)`. This defines the conditional distribution $p_\theta(z_c^t|z_c^{1:t-1}, z_d)$.

Recognition model $q_\phi(z_d, z_c|x)$

- Signal LSTM embedder
  (input size = 1, hidden size = `obs_embedding_dim` = `hidden_size`)
    - This module summarizes the signal $x$ into an embedding vector $e_x \in \mathbb{R}^{\texttt{obs\_embedding\_dim}}$.
    - The embedding vector is taken to be the last hidden state of an LSTM where $x$ is fed as the input sequence.
- Kernel expression LSTM (q)
  (input size = `obs_embedding_dim` + `vocabulary_size`, hidden size = `hidden_size`)
    - This module defines a distribution over kernel expression $q_\phi(z_d|x)$.
    - At each time step $t$, the input is a concatenation of the one-hot encoding of the previous symbol in $\mathcal{V}$ (or a zero vector for the first time step) and $e_x$.
    - At each time step $t$, extract logit probabilities for each element in $\mathcal{V}$ or the end-of-sequence symbol `EOS` from the hidden state using a "Kernel expression extractor (q)" (`Linear(hidden_size, vocabulary_size + 1)`). This defines the conditional distribution $q_\phi(z_d^t|z_d^{1:t-1}, x)$.
    - The full distribution over kernel expression is an autoregressive distribution $\prod_t q_\phi(z_d^t|z_d^{1:t-1}, x)$ whose length is determined by `EOS`.

- Kernel expression LSTM embedder (q)
  (input size = `vocabulary_size`, hidden size = `hidden_size`)
  - This module summarizes the kernel expression $z_d$ into an embedding vector $e_{z_d} \in \mathbb{R}^{\texttt{hidden\_size}}$.
  - The embedding vector $e_{z_d}$ is taken to be the last hidden state of an LSTM where $z_d$ is fed as the input sequence.
- Kernel parameters LSTM (q)
  (input size = `obs_embedding_dim` + `kernel_params_size` + `hidden_size`, hidden size = `hidden_size`)
  - This module defines a distribution over kernel parameters $q_\phi(z_c|z_d, x)$.
  - At each timestep $t$, the input is a concatenation of the previous kernel parameters $z_c^{t-1} \in \mathbb{R}^{\texttt{kernel\_params\_size}}$ (or zero vector for $t = 1$), the embedding vector $e_{z_d} \in \mathbb{R}^{\texttt{hidden\_size}}$ and the signal embedding vector $e_x \in \mathbb{R}^{\texttt{obs\_embedding\_size}}$.
  - At each timestep $t$, extract mean and standard deviations for each kernel parameter in $\mathcal{K}$ using a "Kernel parameters extractor (q)" `Linear(hidden_size, 2 * kernel_params_size)`. This defines the conditional distribution $q_\phi(z_c^t|z_c^{1:t-1}, z_d, x)$.

## B.2 COMPOSITIONAL SCENE UNDERSTANDING

Fixed parameters

- Number of allowed primitives to learn $P = 5$
- Maximum number of blocks per cell $B_{\max} = 3$
- Number of cells in the $x$-plane = number of cells in the $z$-plane $= N = 2$
- Image resolution $I = [3, 128, 128]$
- Observation embedding size `obs_embedding_size` $= 676$

Recognition model $q_\phi(z_d, z_c|x)$. Components are detailed to match those shown in Fig. 7.

- CNN-based embedding of the image

  ```
  - Conv2d(in_channels=4, out_channels=64, kernel_size=3,
    padding=2)
  - ReLU
  - MaxPool(kernel_size=3, stride=2)
  - Conv2d(64, 128, 3, 1)
  - ReLU
  - MaxPool(3, 2)
  - Conv2d(128, 128, 3, 0)
  - ReLU-Conv2d(128, 4, 3, 0)
  - ReLU
  - MaxPool(2, 2)
  - Flatten
  ```

  *Note, the dimensionality of the output is* `obs_embedding_size` *(in this case,* $= 676$*)*

- Neural network for the distribution over blocks (NN1)

  `Linear(obs_embedding_size, ` $N^2$ ` * (1 + ` $B_{max}$ `))`

- Neural network for the distribution over primitives (NN2)

  `Linear(obs_embedding_size, ` $N^2$ ` * ` $B_{max}$ ` * P)`

- Neural network that outputs the mean for the raw primitive locations (NN3)

  ```
  Linear(obs_embedding_size, N^2 * B_max)
  ```

- Neural network that outputs the standard deviation for the raw primitive locations (NN4)

  ```
  Linear(obs_embedding_size, N^2 * B_max)
  ```

## C  COMPARATIVE EFFICIENCY: WALL-CLOCK TIME AND GPU MEMORY CONSUMPTION

We comprehensively investigate the comparative wall-clock time and GPU memory consumption of HMWS and RWS by comparing the algorithms over a range of matched number of likelihood evaluations. Here, we focus on the scene understanding domain, specifically the model wherein scene parses are inferred with color. We find that HMWS consistently results in more time- and memory-efficient inference, as seen in Tables 1 and 2. The reason for this is that while we match HMWS's $K(N + M)$ with RWS's $S$ likelihood evaluations,

- The actual number of likelihood evaluations of HMWS is $K \cdot L$ which is typically smaller than $K(N + M)$ because we only take $L \leq N + M$ unique memory elements (step 2 of Algorithm 1).

- The loss terms in lines 7–9 of Algorithm 1 all contain only $K \cdot M < K \cdot L \leq K(N + M)$ log probability terms $\{\log p_\theta(z_d^m, z_c^{mk}, x)\}_{m=1,k=1}^{M,K}$, $\{\log q_\phi(z_c^{mk}|z_d^m, x)\}_{m=1,k=1}^{M,K}$ and $M$ terms $\{\log q_\phi(z_d^m|x)\}_{m=1}^M$ which means differentiating these loss terms is proportionately cheaper than for RWS which has $K(N + M)$ log probability terms.

- We draw only $N$ samples from $q_\phi(z_d|x)$ and $K \cdot L$ samples from $q_\phi(z_c|z_d, x)$ in HMWS in comparison to $K \cdot (N + M)$ in RWS.

| $S$ or $K(M + N)$ | HMWS | RWS |
|---|---|---|
| 8 | 67 min | 82 min |
| 18 | 136 min | 196 min |
| 32 | 186 min | 327 min |
| 50 | 390 min | 582 min |
| 72 | 499 min | N/A |

Table 1: Wall-clock comparison (time to 5000 iterations). HMWS is faster in terms of absolute time to reach a fixed number of iterations regardless of the setting of the number of likelihood evaluations. Note that HMWS is even able to run with higher number of samples (e.g, $K(M + N) = 72$), unlike RWS - which fails to run due to computationally prohibitive memory requirements.

| $S$ or $K(M + N)$ | HMWS | RWS |
|---|---|---|
| 8 | 2277 MB | 3467 MB |
| 18 | 4645 MB | 7646 MB |
| 32 | 8028 MB | 13556 MB |
| 50 | 12485 MB | 21155 MB |
| 72 | 17963 MB | N/A |

Table 2: Comparison of GPU memory consumption. HMWS is more memory-efficient than RWS over a range of hyperparameter settings. Note that HMWS is even able to run with higher number of samples (e.g, $K * (M + N) = 72$), unlike RWS - which fails to run due to computationally prohibitive memory requirements.

## D    ABLATION EXPERIMENTS

### D.1    "INTERPOLATING" BETWEEN HMWS AND RWS

We "interpolate" between RWS and HMWS to better understand which components of HMWS are responsible for the performance increase. In particular, we train two more variants of the algorithm,

- "HMWS-" which trains all the components the same way as HMWS except the continuous recognition model $q_\phi(z_c|z_d, x)$ which is trained using the RWS loss. This method has a memory.
- "RWS+" which trains all components the same way as RWS except the continuous recognition model $q_\phi(z_c|z_d, x)$ which is trained using the HMWS loss in (10).

The learning curves for the time series model in Figure 9 indicate that the memory is primarily responsible for the performance increase since HMWS- almost identically matches the performance of HMWS. Including the loss in (10) in RWS training helps learning but alone doesn't achieve HMWS's performance.

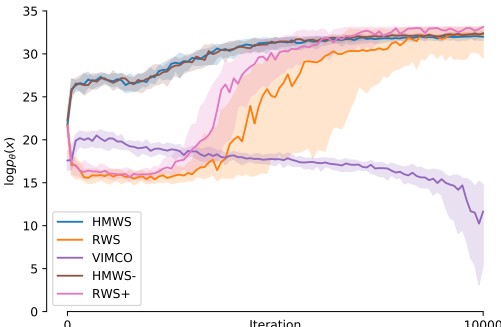

Figure 9: Augmenting the learning curves of HMWS, RWS, VIMCO on the time series domain in Figure 4 (left) by HMWS- and RWS+ highlights the importance of HMWS's memory.

### D.2    SENSITIVITY TO THE DIFFICULTY OF CONTINUOUS INFERENCE

Since both the marginalization and inference of the continuous latent variable $z_c$ is approximated using importance sampling, we expect the performance of HMWS to suffer as the inference task gets more difficult. We empirically confirm this and that RWS and VIMCO suffer equally if not worse (Fig. 10) by arbitrarily adjusting the difficulty of continuous inference by changing the range of the period $p_{\text{PER}_i}$ of periodic kernels (see App. B.1).

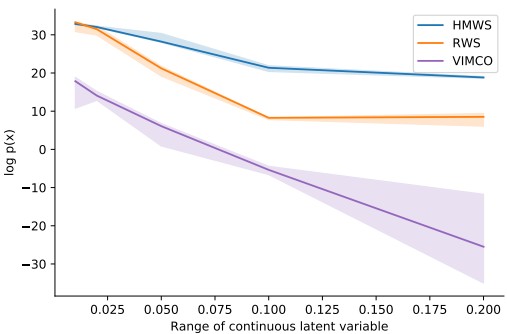

Figure 10: Increasing the difficulty of the continuous inference task results in worse performance in HMWS. Baseline algorithms RWS and VIMCO suffer equally or even more.

### D.3    COMPARING HMWS AND RWS FOR DIFFERENT COMPUTE BUDGETS

We compare HMWS and RWS for a number of different compute budgets (see Fig 11). Here, we keep $K * (M + N) = S$ to match the number of likelihood evaluations of HMWS and RWS, and

keep $K = M = N$ for simplicity. We run a sweep for HMWS with $K = M = N$ in $\{2, 3, 4, 5, 6\}$ and a corresponding sweep for RWS with $S$ in $\{8, 18, 32, 50, 72\}$. We find that HMWS consistently overperforms RWS.

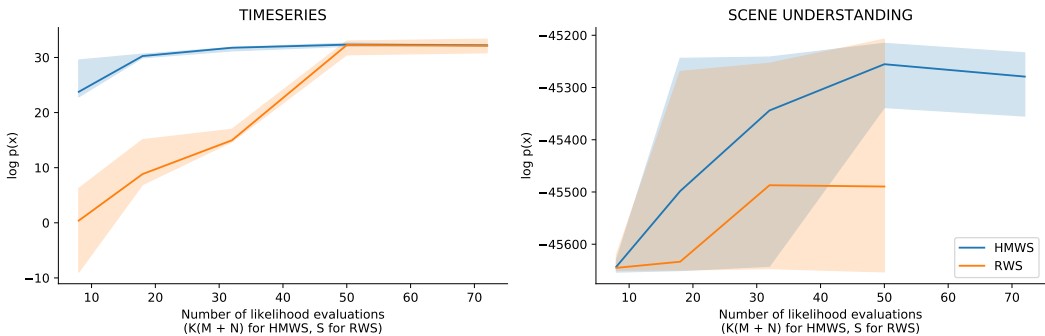

Figure 11: Comparing HMWS and RWS for other compute budgets. Increasing the overall compute by increasing $K$ and keeping $K = M = N$ improves performance while HMWS consistently outperforms RWS with equivalent compute. Note that RWS didn't run for $S = 72$ due to memory constraints.

# E USING VIMCO FOR HYBRID GENERATIVE MODELS

We initially treated the continuous latent variables the same way as the discrete variables. In the reported results, we used reparameterized sampling of the continuous latent variables for training VIMCO which decreases the variance of the estimator of the continuous recognition model gradients. However, this didn't seem to improve the performance significantly.

Since applying VIMCO to hybrid discrete-continuous latent variable settings is not common in the literature, we include a sketch of the derivation of the gradient estimator that we use for completeness.

First, let $Q_\phi(z_d^{1:K}) = \prod_{k=1}^{K} q_\phi(z_d^k)$ and $Q_\phi(z_c^{1:K}|z_d^{1:K}) = \prod_{k=1}^{K} q_\phi(z_c^k|z_d^k)$ be the distributions of all the discrete and continuous randomness. We also drop the dependence on data $x$ to reduce notational clutter. Thus, the gradient we'd like to estimate is

$$\nabla_\phi \mathbb{E}_{Q_\phi(z_d^{1:K})Q_\phi(z_c^{1:K}|z_d^{1:K})}[f(\theta, \phi, z_d^{1:K}, z_c^{1:K})], \tag{26}$$

where $f$ is the term inside of the IWAE objective $f(...) = \log \frac{1}{K} \sum_{k=1}^{K} \frac{p_\theta(z_d^k, z_c^k, x)}{q_\phi(z_d^k, z_c^k|x)}$.

Next, we apply reparameterized sampling of the continuous variables $z_c^{1:K}$. This involves choosing a simple, parameterless random variable $\epsilon \sim s(\epsilon)$ and a reparameterization function $z_c = r(\epsilon, z_d, \phi)$ such that its output has exactly the same distribution sampling from the original recognition network $z_c \sim q_\phi(z_c|z_d)$. Hence, we can rewrite the gradient we'd like to estimate as

$$\nabla_\phi \mathbb{E}_{Q_\phi(z_d^{1:K})S(\epsilon^{1:K})}\left[f(\theta, \phi, z_d^{1:K}, \{r(\epsilon^k, z_d^k, \phi)\}_{k=1}^{K})\right], \tag{27}$$

where $f$ takes in reparameterized samples of the continuous latent variables instead of $z_c^{1:K}$.

Now, we follow a similar chain of reasoning to the one for deriving a score-function gradient estimator

$$\nabla_\phi \mathbb{E}_{Q_\phi(z_d^{1:K})S(\epsilon^{1:K})}\left[f(\theta, \phi, z_d^{1:K}, \{r(\epsilon^k, z_d^k, \phi)\}_{k=1}^{K})\right] \tag{28}$$

$$= \int \left[\nabla_\phi(Q_\phi(z_d^{1:K})S(\epsilon^{1:K}))\right] f(...) + Q_\phi(z_d^{1:K})S(\epsilon^{1:K})\nabla_\phi f(...) \, dz_d^{1:K} \, d\epsilon^{1:K} \tag{29}$$

$$= \int Q_\phi(z_d^{1:K})S(\epsilon^{1:K})(\nabla_\phi \log Q_\phi(z_d^{1:K}))f(...) + Q_\phi(z_d^{1:K})S(\epsilon^{1:K})\nabla_\phi f(...) \, dz_d^{1:K} \, d\epsilon^{1:K} \tag{30}$$

$$= \mathbb{E}\left[\nabla_\phi \log Q_\phi(z_d^{1:K})f(...) + \nabla_\phi f(...)\right]. \tag{31}$$

The VIMCO control variate is applied to the first term $\nabla_\phi \log Q_\phi(z_d^{1:K})f(...)$. Importantly, this term does not contain the density of the continuous variables $Q_\phi(z_c^{1:K}|z_d^{1:K})$. Hence, this term only provides gradients for learning the discrete recognition network unlike if we treated the continuous latent variables the same as the discrete latent variables (in which case a score-function gradient would be used for learning the continuous recognition network as well). The gradient for learning the continuous recognition network instead comes from the second term $\nabla_\phi f(...)$.

