# OpenReview forum: "Hybrid Memoised Wake-Sleep: Approximate Inference at the Discrete-Continuous Interface"
_ICLR.cc/2022/Conference — ICLR 2022 Poster_

### Official Review · Reviewer_pAhD · 2021-10-21

**Correctness:** 4
**Technical Novelty And Significance:** 3
**Empirical Novelty And Significance:** 2
**Recommendation:** 6
**Confidence:** 4

**Main Review:**

Good paper, giving an interesting new method.

My main concerns are around the empirical evaluations and wallclock times.  In particular, I can easily see that memoization in MWS is going to significantly help runtime.  This isn't so clear in the present algorithm, because there's an awful lot more going on (all the steps involving continuous latent variables don't seem to benefit much from memoization).  This concern is further reinforced by comparing algorithms by iterations (it would also be helpful to look at algorithms by runtime).  Could the authors clarify why their method should help runtime (or whether it won't, and its primarily intended to help performance), and include wallclock time comparisons?

I am also surprised by how much better the algorithm performs relative to RWS.  Intuitively, it would seem that HMWS is doing something very similar to RWS, but memoizing the best samples from the approximate posterior over latent variables.  That would seem to help, but it isn't clear to me that it would help _that_ much, as I wouldn't expect the best samples from the approximate posterior to be that different from typical samples.  Can the authors comment? This is especially concerning, given that:
* There is no hyper parameter optimisation (Adam with default hyper parameters).
* There is no discussion of "matching" the RWS and HMWS architectures to ensure a fair comparison (that I could see).

**Summary Of The Paper:**

Memoized wake-sleep is a past method that builds a variational approximate posterior over discrete latent variables by memorising previously drawn samples.  However, memorised wake-sleep can currently be used only on models that are purely discrete.  This paper extends memoized wake-sleep by providing a mechanism for using VI to integrate over continuous latent variables.

**Summary Of The Review:**

Good paper, presenting an interesting new method.  A few concerns as regards evaluations + performance tradeoffs.

---

> ### Author Response · Authors · 2021-11-13
> **Official Response to Reviewer pAhD**
>
> Thank you for reviewing our paper, finding the new method interesting, and the insightful questions which point to ways we can improve our paper.
>
> Regarding your main concern about wall-clock times, we actually include a comparison between HMWS and RWS in Appendix C (in the [originally submitted pdf](https://openreview.net/pdf?id=auOPcdAcoy)) in which we demonstrate that HMWS is consistently faster than RWS. We discuss the way we match the algorithms at the start of section 4. We opt to match based on the number of likelihood evaluations since it is often the most expensive operation, $K \cdot (N + M)$ for HMWS and $S$ for RWS. As a result, HMWS has a considerable wall-clock time advantage of over RWS because
> - The actual number of likelihood evaluations of HMWS is $K \cdot L$ which is typically smaller than $K \cdot (N + M)$ because we only take $L$ unique memory elements (step 2 of Algorithm 1).
> - The loss terms in lines 7-9 of Algorithm 1 all contain only $K \cdot M  < K \cdot L \leq K (N + M)$ log probability terms of the generative model and the recognition model which means differentiating these loss terms is proportionately cheaper than for RWS which has $K (N + M)$ log probability terms.
> - We draw only $N$ samples from $q_\phi(z_d | x)$ and $K\cdot L$ samples from $q_\phi(z_c |z_d, x)$ in HMWS in comparison to $K \cdot (N + M)$ in RWS.
> In the [updated submission pdf](https://openreview.net/references/pdf?id=2hP74-cNFe), we have augmented our discussion of wall-clock time in Appendix C accordingly and point to it explicitly from the paragraph at the start of section 4.
>
> Regarding the comparison of algorithms by runtime instead of by iterations, since HMWS is faster than RWS in runtime per iteration as well as per iteration, it will be even faster in terms of just runtime.
>
> Regarding the underlying reason for HMWS outperforming RWS, we hypothesize that at the start of training while there can be few samples from $q_\phi(z_d | x)$ that are good, this will only weakly benefit training through the corresponding gradients in learning iteration for RWS while for HMWS the good samples will be memoized which can affect training over more learning iterations. This is also the basis of the advantage of MWS over RWS for purely discrete latent variable models. In [response to Reviewer FY4y](https://openreview.net/forum?id=auOPcdAcoy&noteId=H6Mm2wMx1p), we ran additional experiments to further understand the benefit of HWMS over RWS. Reiterating the most relevant part here, in Appendix D.1 of the [updated pdf](https://openreview.net/references/pdf?id=2hP74-cNFe), we “interpolate” between RWS and HMWS to better understand which components of HMWS are responsible for the performance increase. In particular, we train two more variants of the algorithm,
>
> - `HMWS-` which trains all the components the same way as HMWS except the continuous recognition model $q_\phi(z_c | z_d, x)$ which is trained using the RWS loss. This method has a memory.
> - `RWS+` which trains all components the same way as RWS except the continuous recognition model $q_\phi(z_c | z_d, x)$ which is trained using the HMWS loss in eq. 10.
>
> The learning curves indicate that the memory is primarily responsible for the performance increase since `HMWS-` almost identically matches the performance of HMWS. Including the loss in eq. 10 in RWS training helps learning but alone doesn’t achieve HMWS’s performance.
>
> Regarding the lack of hyperparameter optimization, our rationale is that both methods boil down to computing different loss terms for updating parameters of the generative model and the recognition model based on the exact same samples (from $q$) and log probabilities, with the terms just being differently composed for HMWS, and as a result any advantage or difference in performance should not come from tweaking optimizer parameters. Moreover, in comparing different algorithms we use the exact same generative model and recognition model, which further isolates the comparison to the algorithmic aspects of the problem.
>
> Regarding the novelty and significance of our method, we’d like to emphasize that conceptually simple contributions are 1) technically very challenging to implement and 2) can make big contributions to performance as is the case here.

---

> > ### Comment · Reviewer_pAhD · 2021-11-29
> > **Response**
> >
> > Thanks!  That's great.  I would quite like to increase the score to 7.  But for some reason they've banned 7 this year, and we can only have 6 or 8...

---

### Official Review · Reviewer_G5jQ · 2021-10-25

**Correctness:** 4
**Technical Novelty And Significance:** 3
**Empirical Novelty And Significance:** 3
**Recommendation:** 8
**Confidence:** 3

**Main Review:**

**Strengths**

* The paper deals with a relevant issue - how to extend an existing approach to hybrid discrete-continuous graphical models.
* The exposition is clear. Related work and background are explained well. It is made easy to see where the authors' work fits in to previous work.
* The proposed method is principled and based on a well-established method from approximate inference (importance sampling). There are no ad-hoc terms or deviations from theory.
* Experiments are favorable towards the approach and show that it is better suited to program synthesis than other applicable methods.

**Weaknesses**

* Somewhat weak in novelty.

**Other questions/comments**

* In 2.1 you say:

> If we try to use the same approach for hybrid discrete-continuous latent variable models, all proposed continuous values are will be unique and the posterior approximation will collapse onto the MAP estimate.

I don't immediately see why this is the case. Is this an empirical observation or does it follow from theory? Also there is a typo in that sentence.

* In theory we could approximately integrate out the continuous variables using other methods. Did you consider comparing to a simple MC estimate instead of importance sampling (maybe even with a single sample, as done in VAEs)? Or is there a reason why such a comparison is not possible?



**Summary Of The Paper:**

The paper proposes an extension of memoised wake-sleep which allows applications to hybrid discrete-continuous graphical models. The approach is directed towards Bayesian program synthesis.

Specifically, the memoised wake-sleep algorithm requires computing the joint probabiltiy of a discrete latent variable and an observation, which would require integrating out any continuous latents. This paper proposes to solve that problem using importance sampling.

**Summary Of The Review:**

I think the paper is good and recommend accepting.

The main weakness I see is that the contribution seems to be limited to using importance sampling for marginalizing out continuous variables. The actual presence of continuous variables can be seen in the Gaussian mixture models experiment of the memoised wake-sleep paper [1], where it was possible to treat them analytically. Of course, the general case doesn't allow for that and therefore the paper is still making a valuable contribution by investigating what to do in the general setting.

Further, the paper is specifically aimed at settings where discrete variables arise in a programmatic context and impact control flow. The experiments show that the proposed approach has an edge over other general methods in this setting.

[1] Hewitt et al., "Learning to learn generative programs with Memoised Wake-Sleep", https://arxiv.org/pdf/2007.03132.pdf

---

> ### Author Response · Authors · 2021-11-13
> **Official Response to Reviewer G5jQ**
>
> Thank you for carefully reviewing our paper and appreciating its relevance, exposition, technical correctness and the experimental support.
>
> Regarding novelty, we’d like to emphasize that while using importance sampling to marginalize continuous variables and approximate the continuous posterior is conceptually simple, such contributions can be 1) technically challenging to implement and 2) can make big contributions to performance as is the case here.
>
> Regarding other methods of approximate marginalization, we have considered using MCMC methods as well as stochastic VI methods which involve inner loop sampling and optimization respectively for each batch element. While these methods could turn out to be more accurate than importance sampling, the inability to parallelize the inner-loop sampling/optimization led to an impractically slow learning. Using vanilla MC estimation is not directly applicable to marginalizing $z_c$ $p_\theta(z_d, x) = \int p_\theta(z_d, z_c, x) \,\mathrm dz_c$ as it is only used for approximating expectations of the form $\mathbb E_\pi(z)[f(z)]$ which assumes we know how to evaluate the density $\pi(z)$. Indeed, importance sampling based marginalization is derived by reformulating the marginalization as a MC estimation of a weighted function. Moreover it is not straightforward to use MC for approximating the continuous posterior $p_\theta(z_c | z_d, x)$ which is used for deriving the loss for the generative model (eq 7) and the continuous recognition model (eq 10).
>
> Regarding our comment about direct inapplicability of MWS to continuous latent variables in 2.1, this is an empirical observation that is related to the theoretical derivation of MWS ([Hewitt et al., 2020](https://arxiv.org/abs/2007.03132)) - the KL divergence in their section 3 (just above the box) is between a memory $Q_i(z)$ and the posterior $p_\theta(z | x)$ is ill-defined because $Q_i$ and $p_\theta(z | x)$ have different supports ($Q_i$ has discrete support, while $p_\theta(z | x)$ can also have continuous latent variables).

---

> > ### Comment · Reviewer_G5jQ · 2021-11-30
> > **Thank you**
> >
> > Thank you for elaborating on the few minor issues I had.
> >
> > I'm staying by my original score.

---

### Official Review · Reviewer_FY4y · 2021-11-02

**Correctness:** 3
**Technical Novelty And Significance:** 3
**Empirical Novelty And Significance:** 2
**Recommendation:** 6
**Confidence:** 4

**Main Review:**

Strengths:
1. The paper is clearly written with good motivation and methods sections. The rigorous and concise notations convey a lot of information. The diagram in Fig 5 is less clear but many details are given in the appendix.
2. The method is technically sound and involves a non-trivial application of importance sampling, which is of value.
3. The experiments are reasonably complicated (but see below) and serve a good purpose demonstrating the potential of this method.

Weaknesses:
1. My main concern is the significance of this paper. The applications are relevant to comp. cogsci community, but the authors did not discuss much of the cognitive link or compare it with any cognitive/behavioural data. As a machine learning paper, I do not find the comparison with VIMCO and RWS to be sufficient. Further, although the two problems in the experiments involve reasonably complicated generative models, these models are not the main contributions of the paper, and they are arguably low-dimensional in latent space. For example, in the second experiment, there are only 4 discrete locations among other latent variables. Again, as a machine learning paper, the question of scalability is not well addressed. As a simple test, is it possible to learn a flexible parametric likelihood (Gaussian with NN mean) for experiment 2, rather than using a differentiable renderer?
2. The fact that the problems are relatively low-D raises two further questions: how does this method scale to larger latent dimensions? How does this method compare with ablations of this method? As far as I understand, the gap between RWS and the proposed method include a) memory; b) importance sampling to maintain extra dependency between $z_d$ and $z_c$. Either of these can be augmented to RWS for improvement. How do these two contribute to performance? I am not asking the authors to "just compare more baselines", but to conduct a more careful comparison between contributions of various components in their model.
3. Importance sampling has a proveably large variance. How does this method not suffer from this problem? Is it because of the sleep-phase updates? I hope the authors could address this problem.

Other suggestions
1. On the algorithmic side, I'd like the authors to expand a bit more on the paragraph before experiment 4, which I think touches on important upside (mentioned dependence between $z_c$ and $z_d$) and downside: for model where the induced true posterior dependency factories the other way as $p(z_d|z_c,x)p(z_c|x)$, the proposed method may suffer.
2. In the explanation before equation (10), the authors mentioned that "We want to minimize the average of KL". Why does this make sense? The authors could include a derivataion starting from the joint KL
$$
KL[p(z_c,z_d|x) || q(z_c, z_d|x)] = \int  p(z_c,z_d|x) \log \frac{p(z_c,z_d|x)}{q(z_c,z_d|x)} = \int p(z_c|z_d,x)  p(z_d|x) \log \frac{p(z_d|x)}{q(z_d|x)}\frac{p(z_c|z_d,x)}{q(z_c|z_d,x)}
$$
This will then expose another assumption that the aforementioned expectation is in fact taken over the approximate $q(z_d|x)$. This is also related to the expectation propagation method.
3. For the runtime comparison, I'd like the authors to also show the memory cost comparison and be explicit about the tradeoff.

**Summary Of The Paper:**

The paper extends the previous memoized wake-sleep method to train complex generative models with discrete/structural and continuous latent variables. The method incorporates an assumption on the conditional dependence between the discrete and continuous latent variables and essentially employs a clever importance sampling procedure to capture the dependence. The experiments show applications to spatial and temporal compositional models where the proposed method outperforms VIMCO and reweighted wake-sleep.

**Summary Of The Review:**

In general, I think the method is an interesting and valuable contribution, I also enjoyed the precise notations the authors adopted. But either the impact/scalability of the method or the machine learning power needs to be more clearly demonstrated for acceptance.

---

> ### Author Response · Authors · 2021-11-13
> **Official Response to Reviewer FY4y - PART 1**
>
> Thank you for reviewing our paper and bringing up its strengths such as the motivation, the technical soundness and complexity, the clarity of writing, and the complexity of the experiments. We also appreciate the concerns about the impact and scalability and the detailed suggestions about how to improve our paper in this direction. We address these points below.
>
> Regarding the significance of the paper (addressing weakness #1), we see our method as enabling and improving unsupervised learning of generative models which combine 1) the compositionality and interpretability afforded by discrete latent variables and 2) the flexibility afforded by the ability to train neural networks within the generative model (like the LSTM in the timeseries; we can also potentially include neural networks in the generative model of scenes) as well as the ability of having continuous latent variables. As you point out, this class of models is currently not as “scalable” as, for example, models with highly unconstrained neural network likelihoods (Gaussian with NN mean) like VAEs and GANs, especially if “scalability” refers to sampling / reconstruction quality. On the flip side, VAE/GAN models typically have highly uninterpretable latent variables. Algorithms such as HMWS enable learning flexible generative models with neural networks while still being able to inject inductive bias in the form of symbolic latent variables if required.
>
> Moreover, while our models don’t deal with as high-dimensional latent variables as one would encounter in uninterpretable latent variable models (like VAEs and GANs), we’d like to point out that discrete search space is exponential both in the time series domain ($T^V$ where $T=20$ is maximum number of timesteps and $V=11$ is the vocabulary size) and the scene understanding domain (  $(P^{B_max})^{(N^2)}$ using the notation introduced in section 4.2). The added interpretability from having discrete latent variables comes at the price of much more difficult inference as evidenced by the weak performance of the baseline algorithms (RWS, VIMCO).
>
> Regarding the scalability and ablations of HMWS (addressing weakness #2), we appreciate the insightful experiment suggestions which are not “just compare more baselines”. We have [updated our submission pdf](https://openreview.net/references/pdf?id=2hP74-cNFe#page=16) to include an ablation experiment in Appendix D.1 in which we “interpolate” between RWS and HMWS to better understand which components of HMWS are responsible for the performance increase. In particular, we train two more variants of the algorithm,
>
> - `HMWS-` which trains all the components the same way as HMWS except the continuous recognition model $q_\phi(z_c | z_d, x)$ which is trained using the RWS loss. This method has a memory.
> - `RWS+` which trains all components the same way as RWS except the continuous recognition model $q_\phi(z_c | z_d, x)$ which is trained using the HMWS loss in eq. 10.
>
> The learning curves indicate that the memory is primarily responsible for the performance increase since `HMWS-` almost identically matches the performance of HMWS. Including the loss in eq. 10 in RWS training helps learning but alone doesn’t achieve HMWS’s performance.
>
> Regarding the variance of importance sampling (addressing weakness #3), HMWS suffers from this problem---the harder the continuous inference task becomes, the worse the method performs. However, RWS and VIMCO suffer equally if not worse. In Appendix D.2 of [our updated submission pdf](https://openreview.net/references/pdf?id=2hP74-cNFe#page=16), we include an extra ablation experiment to show this. Alternatively, we could also use more powerful inference algorithms such as MCMC. However, this comes at the price of slower runtimes.
>
> (continued at the [Official Response to Reviewer FY4y - PART 2](https://openreview.net/forum?id=auOPcdAcoy&noteId=7JylgEb82D))

---

> ### Author Response · Authors · 2021-11-13
> **Official Response to Reviewer FY4y - PART 2**
>
> (continued from the [Official Response to Reviewer FY4y - PART 1](https://openreview.net/forum?id=auOPcdAcoy&noteId=H6Mm2wMx1p))
>
> Regarding the factorization of latent variables (other suggestion #1), we want to emphasize that in most situations we would like the $q_\phi(z_d | x) q_\phi(z_c | z_d, x)$ factorization as we want a symbolic scaffolding first and continuous refinements second. Moreover, since the posterior $p_\theta(z_d, z_c | x)$ can be factorized either as $p_\theta(z_d | x)p_\theta(z_c | z_d, x)$ or as $p_\theta(z_c | x)p_\theta(z_d | z_c, x)$, our recognition model $q_\phi(z_d | x)q_\phi(z_c | z_d, x)$ can always approximate the posterior exactly. The only scenario in which this is impossible is if the $p_\theta(z_c | x)p_\theta(z_d | z_c, x)$ factorization is forced. This can happen, for example when the existence of discrete latent variables depends on the continuous latent variables, as illustrated in the following simple probabilistic program
> ```
> zc = sample(Normal(0, 1))
> if zc > 0:
> 	zd1 = sample(Categorical([1/3, 1/3, 1/3]))
> 	zd2 = sample(Bernoulli(0.5))
> 	zd = [zd1, zd2]
> else:
> 	zd1 = sample(Categorical([1/4, 1/4, 1/4, 1/4]))
> 	zd = [zd1]
> x = observe(Normal(f(zd), 1))
> ```
> In this case, we cannot use $q_\phi(z_d | x)$ because it is unknown whether $z_d$ has one or two elements.
>
> Regarding the derivation of equation 10 (other suggestion #2), there are many ways for justifying a loss for training an amortized recognition model $q_\phi(z_c | z_d, x)$ the most general of which is $\int w(z_d, x) \text{divergence}(p_\theta(z_c | z_d, x), q_\phi(z_c | z_d, x) \,\mathrm dz_d \mathrm dx$ where $w$ is a positive weighting function of how important it is to be able to do good inference for $(z_d, x)$. Equation 10 is an instance of this where $w(z_d, x) = $ empirical data distribution * empirical distribution based on memory elements given $x$, which reflects the requirement for $q_\phi(z_c | z_d, x)$ approximating the posterior well for $x$ coming from the data distribution and $z_d$ coming from the memory. As you suggest, it is equally valid to derive a loss for training $q_\phi(z_c | z_d, x)$ starting from the joint KL which simplifies to $\mathbb E_{\text{data distribution}}[\mathbb E_{p_\theta(z_d | x)} [ KL(p_\theta(z_c | z_d, x) || q_\phi(z_c | z_d, x)) ]] + \text{const}$. This results in the same loss as in equation 10, except $w(z_d, x)$ is empirical data distribution * $p_\theta(z_d | x)$.
>
> Regarding memory cost (other suggestion #3), we add Table 2 in [Appendix C in the updated pdf](https://openreview.net/references/pdf?id=2hP74-cNFe#page=15) which shows that HMWS is both faster and more memory efficient than RWS. This is because of our conservative matching of HMWS’s number of particles ($K$), number of proposals ($N$) and memory size ($M$) with RWS’s number of particles ($S$): $S = (N + M) K$. While this matches the number of likelihood evaluations, HMWS ends up being much cheaper in other respects.

---

### Author Response · Authors · 2021-11-19
**Do you have any further questions?**

Dear reviewers, thank you so much for your time reviewing our paper so far. In light of our responses, are there any points we could help further clarify or address?

---

> ### Author Response · Authors · 2021-11-22
> **An additional experiment comparing HMWS and RWS for different compute budgets**
>
> We additionally updated [our submission](https://openreview.net/references/pdf?id=BRWgYA1pcO) to include a comparison between HMWS and RWS for different compute budgets in Appendix D.3. We keep $K(M + N) = S$ to match the number of likelihood evaluations of HMWS and RWS, and keep $K = M = N$ for HMWS. We run a sweep for HMWS with $K = N = N \in \{2, 3, 4, 5, 6\}$ and corresponding sweep for RWS with $S \in \{8, 18, 32, 50, 72\}$ . We find that HMWS consistently outperforms RWS.

---

### Decision · Program_Chairs · 2022-01-20

**Decision:**

Accept (Poster)

**Comment:**

The authors propose a method called Hybrid Memoised Wake-Sleep (HMWS) for training models with both discrete and continuous latent variables efficiently using amortized inference. They extend Memoised Wake-Sleep (MWS), which can only handle discrete latent variables, to discrete-continuous systems by using importance sampling to approximately marginalize out the continuous variables and then applying MWS to the discrete variables.

This is well motivated and well written paper. The method is novel, clearly described, and evaluated in two fairly different interesting settings. However, while the empirical evaluation was considerably strengthened by the ablation studies and other experiments included in response to the reviewers, it is still on the weak side. The main issues are the relatively low-dimensional latent spaces and insufficiently tuned baselines. As one reviewer pointed out, importance sampling is unlikely to scale well to high dimensions, so exploring this aspect experimentally would strengthen the paper. The use of default Adam parameters for all methods and models substantially undermines the results and might at least in part explain the underwhelming baseline performance. For example, VIMCO has been successfully used to train a discrete/continuous model of seemingly comparable complexity in [1] and yet here it seems to fail completely. To shed some light on this, the authors might want to explain how they used VIMCO and whether their approach was substantially different from the one from [1].

Finally, it at least somewhat misleading to claim, as is done e.g. in Sec. 4, that HMWS is more memory efficient than the baselines, when unlike them, it needs to store M discrete latent configurations per training case. Please make this claim more precise to avoid potential confusion.

[1] Variational Memory Addressing in Generative Models, Bornschein et al., NIPS 2017